# Strategies to Improve Adherence to Dietary Weight Loss Interventions in Research and Real-World Settings

**DOI:** 10.3390/bs7030044

**Published:** 2017-07-11

**Authors:** Alice A. Gibson, Amanda Sainsbury

**Affiliations:** The Boden Institute of Obesity, Nutrition, Exercise & Eating Disorders, Sydney Medical School, Charles Perkins Centre, The University of Sydney, Camperdown, NSW 2006, Australia; amanda.salis@sydney.edu.au

**Keywords:** adherence, obesity, diet-reducing, appetite

## Abstract

Dietary interventions are the cornerstone of obesity treatment. The optimal dietary approach to weight loss is a hotly debated topic among health professionals and the lay public alike. An emerging body of evidence suggests that a higher level of adherence to a diet, regardless of the type of diet, is an important factor in weight loss success over the short and long term. Key strategies to improve adherence include designing dietary weight loss interventions (such as ketogenic diets) that help to control the increased drive to eat that accompanies weight loss, tailoring dietary interventions to a person’s dietary preferences (and nutritional requirements), and promoting self-monitoring of food intake. The aim of this paper is to examine these strategies, which can be used to improve adherence and thereby increase the success of dietary weight loss interventions.

## 1. Introduction

The obesity epidemic is one of the greatest public health challenges globally. In Australia, obesity has affected every facet of our population, with almost two thirds (63.4%) of adults now considered overweight or obese [1]. Obesity places an enormous burden on society, costing an estimated $8.6 billion dollars annually in Australia alone [2]. Clinicians are at the forefront of the obesity epidemic, often with limited time and resources. In order for clinicians to tackle the obesity epidemic, they need evidenced-based treatments that are practical, affordable and feasible to implement in real word settings.

The core principle of any obesity treatment is that it must shift the balance between energy intake and energy expenditure. Diet-induced weight loss, or ‘dieting’ as it is usually referred to, is the most common approach to weight loss. For a person dieting to lose weight, the energy balance principle dictates that they must restrict their energy intake to below their total energy expenditure in order to induce an energy deficit. However, there are many ways to achieve energy restriction due to the complex interplay of human metabolism, eating behaviours and the food supply, and this has contributed to the seemingly endless array of dieting books, products and programs available. The optimal dietary approach to weight loss is thus a hotly debated topic among health professionals and the lay public alike.

While the debate about the optimal weight loss diet continues, an emerging body of evidence suggests that higher levels of adherence to a diet, regardless of the type of diet, is an important factor in weight loss success [3,4,5,6]. There are numerous factors that may influence an individual’s level of adherence to a dietary intervention, and clinicians will likely need to utilise a range of strategies to promote adherence. This paper will first narratively review the evidence supporting adherence as a key factor in weight loss success. It will then narratively review three key strategies that can be used to improve adherence and thereby increase the success of dietary weight loss interventions. These three strategies to be reviewed here are designing dietary weight loss interventions that help to control the increased drive to eat that accompanies weight loss, tailoring weight loss interventions to a person’s dietary preferences (and nutritional requirements), and promoting self-monitoring of food intake. Although not the only strategies that can be utilised in practice to promote dietary adherence, we selected these specific strategies because they are practical, feasible, and applicable in both research and real-world settings, in diverse clinical populations.

## 2. Adherence is the Key to Weight Loss Success

A diverse body of research supports the idea that dietary adherence—the degree to which an individual ‘sticks’ to a diet—is a more important factor in weight loss success that the ‘type’ of diet an individual is prescribed. Dansinger et al. [3] reported a strong curvilinear association (r = 0.60; *p* < 0.01) between self-reported dietary adherence (on a scale of 1 to 10) and weight loss after following one of four popular weight loss diets (Atkins, Zone, Ornish or Weight Watchers), with no association between diet type and weight loss [3]. The Atkins, Zone, Ornish and Weight Watchers diets represent fairly wide extremes in macronutrient distribution. The Atkins diet is very low in carbohydrate (<20 g per day initially); the Zone diet aims to maintain a macronutrient balance with 40-30-30 per cent of energy coming from carbohydrate, protein and fat, respectively; the Ornish diet is a very low-fat (<10% of energy) vegetarian diet; and Weight Watches uses a ‘points’ food exchange system to reduce energy intake [3]. Similarly, results from the A TO Z weight loss study found that dietary adherence score (calculated by comparing self-reported food intake with the dietary goals of the respective diets) was significantly correlated with weight loss at 12-months within each of the three diets tested in that study (Atkins, Zone and Ornish) [4]. Therefore, in these two studies, it was the level of adherence to a diet, not which type of diet an individual followed, that predicted weight loss success. As the diets tested in these two studies are quite contrasting in their macronutrient distributions, it would suggest that in order to increase the success of weight loss diets, more emphasis should be placed on strategies for increasing adherence, rather than on the macronutrient composition of the diet *per se*.

Adherence has also been shown to be an important predictor of longer-term weight loss success in studies where individuals are all prescribed the same diet. Corral et al. [5] quantified dietary adherence based on metabolised energy intake, which was calculated using doubly labelled water to measure energy expenditure, coupled with dual-energy X-ray absorptiometry to measure change in body energy stores before and after a low energy diet (LED; ≥3350 kJ/day or ≥800 kcal/day). All food was provided to participants, and the composition of the LED was 20–23% of energy from fat, 20–23% of energy from protein, and 56–59% of energy from carbohydrate [5]. People in the highest tertile of adherers were found to consume an average of 2692 ± 309 kJ/day (644 ± 74 kcal/day), in contrast to those in the lowest tertile of adherers, who consumed an average of 6575 ± 138 kJ/day (1573 ± 33 kcal/day) [5]. Consequently, high adherers were more successful, losing weight at twice the rate as low adherers (126.5 ± 7.7 g/day versus 56.9 + 2.7 g/day; *p* < 0.001) [5]. Further, adherence during the weight loss phase predicted weight maintenance at two years, with the high adherers regaining only 50% of the weight that was lost, compared with 99% regain for the low adherers [5]. Thus, identifying poor adherence during the early phases of a weight loss diet might be an important indicator of poorer long-term outcomes. Poor dietary adherence has also recently been shown, through the use of mathematical models predicting energy intake, to be the primary reason for the less than expected weight loss associated with energy-restricted diets [7,8,9,10]. The mathematical models are based on the energy balance principle, which calculates average energy intake over a given period of time by deducting changes in energy stores from measures of energy expenditure. As the models take into account the concomitant reduction in energy expenditure with weight loss, it has been shown that it is fluctuating energy intake (i.e., poor dietary adherence) that is the primary driver of the weight loss plateau that typically occurs around 6-months into a diet [7]. Taken together, the studies highlighted above suggest that whether individuals are given the same diet or different diets, and whether adherence to the diet is determined through self-report, gold standard procedures or mathematical modelling, adherence is strongly associated with weight loss success over the short and long term.

Adherence is surprisingly high with very low energy diets (VLED). VLEDs involve severely restricting energy intake to ≤3350 kJ/day (≤800 kcal/day), which results in substantial and rapid initial weight loss [11]. VLEDs are the most intensive dietary intervention for obesity and the single most effective dietary intervention [12]. The rapid weight loss of VLEDs is very motivating for people and may be one reason they are associated with high adherence. A recent study showed that a person commencing a VLED has an ~80% chance of losing ≥12.5% of their initial body weight, compared to only ~50% for people commencing a diet involving moderate energy restriction [13]. Further, rapid initial weight loss has been shown to be predictive of long term success in maintaining a lower body weight [14,15,16,17]. The use of VLEDs to induce rapid weight loss, in contrast to many of the other weight loss products on the market, is backed by decades of medical research, and VLEDs have been in clinical use for almost 40 years [18,19]. The National Health and Medical Research Council (NHMRC) of Australia and the Dietitians Association of Australia (DAA) clinical practice guidelines for the treatment of overweight and obesity in adults suggest that VLEDs are useful as an intensive medical therapy option and are effective for supporting weight loss in adults with a BMI > 30 kg/m^2^, or with a BMI > 27 kg/m^2^ and obesity-related comorbidities, taking into account the individual situation and when used under medical supervision [20,21]. However, a 2008 survey of Australian dietitians revealed that only 1.5% of respondents reported prescribing a rapid weight loss program to their clients [22]. This is in line with an earlier survey of Australian dietitians, conducted in 2002, in which only 3.2% of respondents reported using a VLED to manage overweight and obesity in their clients [23]. Thus, despite the efficacy of and medical research behind VLEDs, they appear to be underutilised by clinicians, and should be explored further as a means of improving adherence to a weight loss program.

In summary, adherence to a dietary weight loss intervention is strongly associated with weight loss success over the short and long term [3,4,5,6,7,9]. This review will now examine three key strategies to improve adherence to dietary weight loss interventions. These are: designing diets to help reduce the weight loss-induced drive to eat, tailoring the diet to match dietary preferences (whilst meeting nutritional requirements), and promoting self-monitoring of food intake.

## 3. Strategies to Improve Adherence

### 3.1. Reducing the Drive to Eat with Diets that Induce Ketosis

Adherence to dietary weight loss interventions could be improved by strategies that help to control the physical drive to eat that occurs during energy restriction. An increase in the drive to eat in response to energy restriction and weight loss is one of a range of compensatory responses that collectively oppose ongoing weight loss and promote weight regain [24]. An increase in the drive to eat may contribute to the high rate of attrition in weight loss attempts and the inability of most individuals to maintain weight loss [25,26,27,28]. Indeed, the degree of hunger experienced by individuals with overweight or obesity in response to an energy restricted diet has been shown to be a predictor of subsequent weight regain [29], presumably due to increased food intake. However, not all studies have shown an association between changes in hunger and weight regain [30]. The compensatory increase in the drive to eat during weight loss is likely induced by multiple pathways, including alterations in expression of hypothalamic regulators of energy balance [31], as well as adaptive changes in gut function, which alter the concentration of appetite-regulating hormones such as ghrelin [30,32,33], cholecystokinin [30,34,35] and peptide YY [30,34,36]. Thus, designing dietary strategies that ‘block’ the compensatory increase in the drive to eat associated with energy-restricted diets represents a key target to improve adherence.

One dietary strategy that is frequently reported to suppress the drive to eat is diets that induce ketosis. Ketosis is a coordinated metabolic response to a low carbohydrate intake, resulting in an increased circulating concentration of ketones bodies or ‘ketones’ (β-hydroxybutyrte, aceotoacetate and acetone) that are produced by the liver from β-oxidation of free fatty acids. Suppression in the drive to eat associated with ketosis is thought to be a key factor in the efficacy of VLEDs, by allowing individuals to adhere to a severe restriction of energy intake and lose weight rapidly, without a compensatory increase in hunger [18,19,37,38,39,40]. Indeed, our recent systematic review and meta-analysis showed that individuals are indeed slightly, but significantly, less hungry and exhibit significantly greater fullness/satiety when adhering to a VLED, than when they are in energy balance before the diet [41]. Another type of diet that induces ketosis is ketogenic low carbohydrate diets (KLCDs), which severely restrict dietary carbohydrate intake but allow ad libitum consumption of protein and fat. However, although both KLCDs and VLEDs induce ketosis, KLCDs can result in several-fold higher circulating levels of ketones compared to VLEDs, as KLCDs limit carbohydrate intake to less than 20 g/day, while VLEDs typically provide at least 50–60 g/day [42,43]. The reason why KLCDs induce weight loss is attributed to suppression of the drive to eat, resulting in a spontaneous decrease in energy intake [42,44,45,46]. In keeping with this, our review also found that individuals adhering to KLCDs report feeling significantly less hungry and report feeling a significantly reduced desire to eat compared with baseline measures; albeit due to inclusion of only three KLCDs in the review the evidence should be considered carefully [41]. As the absolute changes shown in the meta-analysis may be considered small, and for some aspects of appetite were not significant, the clinical relevance of these findings to improve adherence is in the clear lack of an *increase* in the drive the eat, particularly for the VLEDs. Thus, the compensatory increase in the drive to eat that typically occurs during energy restriction [24,25,26,27,28] appears to be ‘blocked’ by VLEDs or KLCDs, aiding adherence to the energy restriction required for weight loss via VLEDs or KLCDs. For further discussion of the evidence for possible reasons underpinning the hunger suppressing effects of ketosis and ketogenic diets, see our recent review [41].

Determining whether ketosis per se is involved in the mechanism of hunger suppression during ketogenic diets, and if so, what level of ketosis is needed to result in a suppressed drive to eat, could lead to the development of novel weight management strategies. After all, VLEDs are only intended for short term use [47], and KLCDs, while efficacious in inducing weight loss in the short term, cannot be adhered to in the long term for most individuals [48]. Further, in order for a person to remain in ketosis, they must sustain a severely restricted carbohydrate intake, which involves the elimination, or extremely limited intake, of whole food groups that are beneficial to health (e.g., wholegrains, legumes, fruits, dairy and starchy vegetables). However, if ketosis were found to be involved in the mechanism underlying a suppressed drive to eat, ketosis could potentially be mimicked via the administration of synthetic ketones, thereby aiding adherence via increased hunger control while allowing consumption of a diet that is more aligned with healthy eating guidelines. Alternatively, if the level of ketosis needed for suppressing the drive to eat was found to be low, a diet less restrictive in carbohydrate could be followed, which might be easier for individuals to adhere to and is also more nutritionally sound. In our systematic review and meta-analysis, it was not possible to determine whether ketosis had a dose-dependent effect on the drive to eat, as differences in ketosis levels between the studies were insufficient for performance of a meaningful meta-regression analysis (all studies in the review reported an average circulating ketone level of about 0.5 mM). Elucidating what level of ketosis is required to suppress the drive to eat in response to energy restriction would have important implications for clinical practice. For example, it could allow for the design of KLCDs that are more aligned with dietary guidelines through the inclusion of some healthful carbohydrate containing foods. In turn, the diet would be more adaptable to a person’s usual diet, and therefore more likely to be adhered to in the long term [6,49].

In contrast to inducing ketosis via a carbohydrate restricted diet, a novel approach to weight management could be to mimic ketosis through exogenous administration of synthetic ketones. Synthetic ketones administered orally as part of a meal replacement diet have recently been shown to be safe and well tolerated in humans [50]. Although synthetic ketones were developed as an alternative means of mimicking ketosis for the purposes of treating neurological conditions, such as refractory epilepsy, Parkinsons Disease and Alzheimer’s Disease [51,52], artificially mimicking ketosis could potentially allow individuals to reap the benefits of hunger suppression, while adhering to a diet with a more flexible—and thus sustainable and health-promoting—macronutrient distribution. Further investigation would be required to test this possibility.

In summary, while it appears that ketogenic diets do suppress a compensatory increase in the drive to eat in response to energy restriction and weight loss, and that ketosis provides a plausible mechanism underlying the effect, it is not clear what level of ketosis needs to be reached for this effect to occur. Future research should investigate the minimum level of ketosis required to achieve a suppression in the drive to eat, and the level of carbohydrate intake and/or dosage of synthetic ketones required to achieve this, as a potential means of promoting long term adherence to energy restricted diets via increased control of the drive to eat.

### 3.2. Tailoring Diets to Dietary Preferences (Whilst Meeting Nutritional Requirements)

While controlling the drive to eat may be a key target for improving adherence to dietary weight loss interventions, physical hunger is not the only reason people eat (or drink) (see [53] for further discussion of non-hunger-related factors affecting adherence). Level of adherence may also be influenced by how different a dietary intervention is from a person’s usual diet. Indeed, one study which used a Mediterranean diet intervention found that adherence to the Mediterranean diet at one and four years follow-up (as assessed by a score of ≥7 in the Mediterranean dietary score) was associated with how similar their usual diet was to the Mediterranean diet at baseline [49]. Another study showed that adherence to one of four diets that were high or average in protein, or high or low in fat (2 × 2 factorial design), was better in those who were randomly assigned to a diet which reflected the macronutrient profile of their baseline diet [6]. Therefore, adherence to a dietary intervention is likely influenced by how ‘different’ the dietary intervention is from an individual’s usual diet. This is probably why individuals have difficulty adhering to severely carbohydrate-restricted diets over the long term [6,48], as carbohydrates are a major contributor to energy in most people’s diets [54]. In summary, a dietary intervention that is flexible and can be individualised and adapted according to a person’s dietary preferences may lead to better adherence to dietary prescriptions.

As well as individualisation according to a person’s usual diet and dietary preferences, dietary weight loss interventions should take into account a person’s nutritional requirements and be nutritionally sound. Notably, protein is a nutrient that is particularly important during weight loss for promoting satiety (which may also improve adherence by controlling the drive to eat), as well as helping to prevent loss of fat-free mass [55,56,57,58]. However, there are currently no practical resources (or information in clinical practice guidelines) on how to tailor dietary interventions that allow dietary preferences to be taken into account while also achieving adequate protein intake for individuals with varying requirements. Further, in published studies of dietary weight loss interventions in which protein intake has been individualised to requirements, there is limited information about how the diet was designed and how the diet is typically provided to the participants, be it as specially formulated meal replacements [59,60], or pre-prepared meals [42,61,62]. The diets used in these research studies, while important for establishing efficacy, are difficult to directly translate into clinical practice in real world settings, where individuals purchase commercial meal replacement products or are responsible for purchasing and preparing their own food. In summary, there is a need for practical clinical guidance on how to design dietary interventions that can be tailored to individual dietary preferences and dietary requirements (particularly protein) to improve adherence, that are also applicable in real world settings.

To demonstrate how to achieve a flexible and individualised diet according to a person’s dietary preferences and nutritional requirements on a practical level, we recently published a modelling study which outlines the design process and underlying rationale for the dietary interventions of a clinical trial of fast versus slow weight loss (The TEMPO Diet Trial, ACTRN: 12612000651886) [63]. For the slow weight loss arm of the TEMPO diet trial, that paper demonstrated the feasibility of designing a food-based, moderately energy-restricted diet that can be individualized to a person’s dietary preferences, while still aligning with the Australian Dietary Guidelines and containing adequate protein for women of various sizes. Although the TEMPO Diet Trial is more of an efficacy study than an effectiveness study, due to being conducted in a research setting with a narrowly defined population, it was designed to maximize clinical utility by drawing on existing resources and clinical practice guidelines [20,64]. By demonstrating how to operationalize the Australian Dietary Guidelines for weight loss, this paper provides clinicians with a practical, affordable and feasible intervention that can be adapted and implemented into research or real-world settings.

Taking into account nutritional requirements is particularly important with VLEDs, as they involve the use of specially formulated meal replacement products that replace all usual food intake (except low energy vegetables or broth). The nutritional formulation and cost of meal replacement products used for VLEDs can vary widely, as demonstrated by our recent survey of products available in Australia (several of which are available internationally) [65]. However, as we have shown in our recent papers [63,65], it is possible to tailor VLEDs to suit individuals’ nutrient requirements based on age- and sex-appropriate dietary recommendations [66]. While this does not make the VLED closer to a person’s usual diet, because a VLED is very unlike a person’s usual diet due to the use of meal replacement products, it does make the diet closer to a person’s nutritional requirements. Tailoring to meet individual protein requirements may be particularly important for promoting adherence by helping to control the drive to eat [55,56,57,58]. Tailoring VLEDs to meet nutritional requirements may also help to reduce complications. Although modern VLEDs are accepted as being safe, there are several potential complications associated with VLEDs. These side effects include, but are not limited to, lethargy, light headedness or dizziness, constipation, menstrual irregularities, gastrointestinal upsets, cold intolerance, dry skin and gallstones [67]. The majority of side effects are generally considered insufficient in magnitude or duration to warrant stopping the diet, and usually resolve upon the reintroduction of food. However, VLEDs should ideally be commenced in consultation with an appropriately qualified health care practitioner, particularly for people with weight-related comorbidities such as diabetes. Ensuring that VLEDs contain adequate fat and fibre levels may help to promote adherence by reducing the risk of complications such as gallstones and constipation—the risk of which is higher with VLEDs than with LEDs [68].

### 3.3. Self-Monitoring of Food Intake

Another potential target for improving adherence to a dietary intervention is recording of food intake. Keeping a food record, or ‘food diary’, is common in research and real-world settings for promoting and measuring adherence to dietary interventions, particularly for weight management [69,70,71]. In a systematic review of self-monitoring in weight management, all 15 studies that focused on dietary self-monitoring (in the form of a paper or electronic food diary) found significant associations between the frequency or consistency of self-monitoring of diet and weight loss [71]. Further, self-monitoring via recording of food intake has been shown to be a strong predictor of dietary change, as well as being a strong predictor of maintenance of dietary change over the long term [72]. However, food records can be subject to large errors, particularly with estimation of portion sizes [73]. A key aspect of any dietary intervention is providing dietary guidance on what and how much (i.e., portion size) to eat. Portion size estimation is difficult, but particularly when individuals are away from home and without access to scales or other portion size estimation aids (such as household measuring cups and spoons). A number of strategies have been developed to help individuals estimate portion sizes more accurately, including using comparison to common objects (such as tennis balls, mobile phones, matchstick boxes) as well as using the hands. For example, ‘a fist’, ‘thumb tip’ and ‘fingertip’ are used to estimate one cup, one tablespoon and one teaspoon, respectively. Although the accuracy of these estimation methods may be challenging in a research setting, efforts are currently underway to address this [74]. However, in a real world setting the value of a person self-monitoring their food intake is more so on increasing a person’s awareness of their food intake, rather than accuracy of portion size estimation. This is reflected in the review mentioned above which found associations with the frequency and/or consistency of reporting and weight loss, not necessary the degree of accuracy. In summary, to promote adherence to dietary interventions, clinicians should encourage individuals to self-monitor their food intake.

## 4. Conclusions

Adherence is an important key to weight loss success, and there are a number of strategies that can be used to improve adherence that are applicable in research or real-world settings. An increased drive to eat is a major contributor to unsuccessful weight loss attempts, and thus it is a key target in improving adherence. Diets which induce ketosis (such as VLEDs or KLCDs) may help to control the increased drive to eat associated with weight loss, but further research is needed on the level of carbohydrate restriction that is required to achieve this. Ensuring that a diet contains adequate protein may also help to prevent an increase in the drive to eat. In addition, a dietary intervention that is tailored to a person’s dietary preferences (whilst still aligning with nutritional recommendations), may also improve adherence. For this reason, government-based dietary guidelines are a very useful tool to use when tailoring a dietary intervention, as they are intended as population approach that are designed to be adapted to different dietary, cultural and cost preferences. Encouraging individuals to self-monitor their food intake has also been shown to improve the success of weight loss attempts and maintaining dietary changes overtime.

As alluded to in our Introduction, strategies that can be used to increase adherence are not limited to those discussed herein. There are numerous other strategies that have the potential influence an individual’s level of adherence to a dietary intervention, and clinicians will likely need to utilise a range of strategies, both dietary and behavioural. For instance, increased dietary fibre intake may help to control the drive to eat [75]. However, further evidence for the effect over the long term, and during weight loss and maintenance in individuals with overweight or obesity is required. As well as dietary factors, other behavioural strategies in addition to self-monitoring, such as meal planning [76] may also help to promote adherence and increase the success of weight loss. Given the emerging body of evidence suggesting that higher levels of adherence to a diet, regardless of the type of diet, is an important factor in weight loss success, research efforts should be focused on increasing the evidence based for strategies to improve adherence.

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
