# Peer review of "Strategies to Improve Adherence to Dietary Weight Loss Interventions in Research and Real-World Settings"

_behavsci, 2017, doi:10.3390/bs7030044_

Round 1

Reviewer 1 Report

Overall comments: This is a very well written and relevant review on one of the most pressing challenges within obesity management – adherence to dietary weight loss interventions. I only have minor comments.

Minor comments:

1. Regarding the usage of VLEDs in obesity management, a sentence should be added regarding safety. How do VLEDs compare to LEDs regarding side effects. For sure, risk of gallstone formation and cholecystectomy is higher (see Johansson et al, 2013 IJO).

2. In lines 139-141, the authors state that: “An  increase  in  the  drive  to  eat  in  response  to  energy  restriction  and  weight  loss  is  one  of  a  range  of  compensatory  responses  that  collectively oppose ongoing weight loss and promote weight regain [24]. The claim that increased appetite after weight loss drives weight regain is too strong and not backed by scientific evidence. Even though it makes physiological sense that increased appetite may contribute to weight regain, evidence linked the two is missing. For example in the landmark paper from Sumithran et al 2011 (NEJM) no correlation was seen between changes in appetite (either subjective appetite feelings or plasma concentrations of appetite-related hormones) with weight loss and weight regain at 1 year follow up.

3. The authors state in lines 155-167 that individuals feel less hungry when  adhering  to  a  VLED or KLCD  than  when  they  are  in  energy balance before the diet. The findings are interesting, but they would be more relevant if there was evidence that participants would feel less hungrier when losing weight with a VLED vs losing the same amount of weight with a non-ketogenic diet. Is there evidence for this, either from RCTs or from one arm longitudinal repeated measures design studies that have also looked into appetite after a period of energy balance and no ketosis?

4. In line 266, there is a misspelling. “Particarly” should read particularly

Author Response

Referee 1:

Overall comments: This is a very well written and relevant review on one of the most pressing challenges within obesity management – adherence to dietary weight loss interventions. I only have minor comments.

Minor comments:

1. Regarding the usage of VLEDs in obesity management, a sentence should be added regarding safety. How do VLEDs compare to LEDs regarding side effects. For sure, risk of gallstone formation and cholecystectomy is higher (see Johansson et al, 2013 IJO).

Our response: We had added discussion on the safety and potential side effects of VLEDs, as well as acknowledging the greater risk of gallstones with VLEDs vs LEDs. This is shown in market up text in the 4th paragraph under section 3.2 Tailoring diets to dietary preferences (whilst meeting nutritional requirements).

2. In lines 139-141, the authors state that: “An  increase  in  the  drive  to  eat  in  response  to  energy  restriction  and  weight  loss  is  one  of  a  range  of  compensatory  responses  that  collectively oppose ongoing weight loss and promote weight regain [24].” The claim that increased appetite after weight loss drives weight regain is too strong and not backed by scientific evidence. Even though it makes physiological sense that increased appetite may contribute to weight regain, evidence linked the two is missing. For example in the landmark paper from Sumithran et al 2011 (NEJM) no correlation was seen between changes in appetite (either subjective appetite feelings or plasma concentrations of appetite-related hormones) with weight loss and weight regain at 1 year follow up.

Our response: We have toned down the strength of our statement, from ‘probably contributes’ contributes to ‘may contribute’. We have also added a reference in support of this statement and acknowledged inconsistencies in the literature in this area. This is shown in marked up text in the 1st paragraph under section 3.1 Reducing the drive to eat with diets that induce ketosis.

3. The authors state in lines 155-167 that individuals feel less hungry when adhering to a VLED or KLCD  than  when  they  are  in  energy balance before the diet. The findings are interesting, but they would be more relevant if there was evidence that participants would feel less hungrier when losing weight with a VLED vs losing the same amount of weight with a non-ketogenic diet. Is there evidence for this, either from RCTs or from one arm longitudinal repeated measures design studies that have also looked into appetite after a period of energy balance and no ketosis?

Our response: These are very valid points. In our systematic review and meta-analysis that we cite in the present review we attempt to address and discuss these points in detail. Of note, among five studies (all VLEDs) which were excluded from the meta-analysis because the baseline assessment of appetite was not conducted under conditions of energy balance (but was instead preceded by an energy-restricted diet), participants were either less hungry or were no hungrier while on the VLED compared with after the moderately energy-restricted diet. This finding provides indirect support that individuals are less hungry on a VLED than when on a moderately energy restricted diet. We only found two studies which compared a ketogenic diet with a non-ketogenic diet. However, the non-ketogenic comparison groups in these two studies were too dissimilar to be combined in a meta-analysis, and only one of these studies (Johnston et al Am J Clin Nutr 2008; 87: 44–55) used participants as their own control. In this study, those on a ketogenic high protein diet were less hungry than those on a non-ketogenic high protein diet. To our knowledge this is the only RCT evidence that individuals are less hungry after similar weight loss when in ketosis than when not in ketosis. In regards to studies which have looked into appetite after a period of energy balance, this was not specifically addressed in our systematic review, but again is discussed in the systematic review.  Specifically, the finding that after a period of re-feeding that abolishes ketosis post-VLED, perceived appetite and circulating levels of ghrelin increased, whereas that of the appetite-suppressing hormone CCK decreased, compared with the levels found at baseline and in energy balance (Chearskul et al. Am J Clin Nutr 2008, Sumithran et al. Eur J Clin Nutr 2013). These findings provide indirect support for the hypothesis that changes in appetite, as well as the effect of changes in circulating levels of appetite regulating hormones that have been shown to accompany diet-induced weight loss, appear to be ‘blocked’ during ketosis.

We believe that a detailed discussion of the above considerations is beyond the scope of the present review. However, we have added a sentence to direct readers to the full text of the systematic review and meta-analysis for further discussion of the evidence of the hunger suppression effects of ketosis and ketogenic diets. This is shown in marked up text of the 2nd paragraph of section 3.1 Reducing the drive to eat with diets that induce ketosis.

4. In line 266, there is a misspelling. “Particarly” should read particularly

Our response: This mistake has been corrected, thank you.

Reviewer 2 Report

This is a succinct and accurate review of the literature in an area of increasing interest. The debate about 'type' of diet and/or food in weight control has now gotten out of hand with exponents of various views holding fast to their beliefs in the presence of conflicting evidence - both ways. The current report brings the argument back to one of 'adherence' - irrespective of diet type; a much more obvious and sensible approach.

I have two minor comments to make:

1. The paper isolates three potential processes for improving adherence. It also hints at other approaches (eg. increased protein use) for making adherence potentially easier. However it does seem to leave out some others (increased fibre intake for example). I am aware that this, and some other approaches (possibly involving exercise), may not have as much supporting evidence as those discussed here. However, it might be nice to list other proposed approaches in the final discussion in the paper and point to the need for further evidence.

2. I caution against the use of 'appetite' and 'appetite' suppression, when what we really mean here is 'hunger'. Appetite, as defined by the Oxford English Dictionary is a learned, psychological drive. Hunger is an unlearned biological drive. The two have two totally different connotations which is often ignored in academic publications  (because medical dictionaries do not make this distinction).  I would like to see this recognised here. 

Author Response

Referee 2:

This is a succinct and accurate review of the literature in an area of increasing interest. The debate about 'type' of diet and/or food in weight control has now gotten out of hand with exponents of various views holding fast to their beliefs in the presence of conflicting evidence  both ways. The current report brings the argument back to one of 'adherence' - irrespective of diet type; a much more obvious and sensible approach.

I have two minor comments to make:

1. The paper isolates three potential processes for improving adherence. It also hints at other approaches (eg. increased protein use) for making adherence potentially easier. However it does seem to leave out some others (increased fibre intake for example). I am aware that this, and some other approaches (possibly involving exercise), may not have as much supporting evidence as those discussed here. However, it might be nice to list other proposed approaches in the final discussion in the paper and point to the need for further evidence.

Our response: We thank the Referee’s for this suggestion. We have added a paragraph to the final section of the paper to highlight that other strategies can be used to improve adherence, and the need for more evidence underpinning these. This is shown in marked up text in the second paragraph of section 4. Conclusions.

2. I caution against the use of 'appetite' and 'appetite' suppression, when what we really mean here is 'hunger'. Appetite, as defined by the Oxford English Dictionary is a learned, psychological drive. Hunger is an unlearned biological drive. The two have two totally different connotations which is often ignored in academic publications  (because medical dictionaries do not make this distinction).  I would like to see this recognised here.

Our response: We agree with the Referee’s distinction and throughout the paper have preferred to use the terminology of ‘drive to eat’ in place of appetite in order to distinguish between physiological appetite sensation such as hunger and psychological drive or ‘hedonic hunger’ as it is also sometimes referred to. In the sections where we had still used the word ‘appetite’ or ‘appetite suppression’ (section 3.1 Reducing the drive to eat with diets that induce ketosis) we have changed this to ‘hunger’ and ‘hunger suppression’ for greater clarity.